# OpenReview forum: "SecTest-Eval: Can LLMs Verify Security Impacts of A Vulnerability?"
_ICLR.cc/2026/Conference — Submitted to ICLR 2026_

### Official Review · Reviewer_Q1vV · 2025-10-31

**Soundness:** 2
**Presentation:** 2
**Contribution:** 1
**Rating:** 2
**Confidence:** 4

**Summary:**

This paper introduces a new benchmark, SecTest-Eval, for evaluating LLMs on the task of generating unit tests for verifying security impacts of vulnerable methods.
This subtly differs from the more common task of generating tests for vulnerable code snippets in that it requires explicit generation of "sensitive statements" that lead to security impacts.
The paper argues that existing benchmarks either focus on project-level test generation which might not isolate the vulnerability test generation capabilities of LLMs (SEC-Bench), or focus on limited security impacts (SEC-Bench, CyberEval).
The proposed benchmark focuses on function/method-level vulnerable code and three security impact categories: confidentiality (reading and printing sensitive information), integrity (modifying sensitive information), and availability (early termination or unresponsiveness).
The function-level vulnerable code snippets (C/C++) are obtained from existing vulnerability detection datasets (PRIMEVUL) which are then filtered using LLMs and manual inspection to 204 instances which contain sensitive statements (balanced by the three categories, 14 vulnerability types).
Finally, the paper finds that 5 LLMs underperform on SecTest-Eval (~56% success rate by the best LLM) and lists the differences in terms of model type (general purpose vs. code LLMs), security impact category, function length, prompting technique, and vulnerability type.

**Strengths:**

- While there are benchmarks that evaluate detection of function-level vulnerabilities, the proposed benchmark is the first to look at the security impacts of these vulnerabilities.
- The experiments section is well-written and easy to understand.

**Weaknesses:**

- The models evaluated in the experiments are LLMs with two simple prompting strategies. The state-of-the-art methods on several Software Engineering tasks are agentic frameworks such as SWE-agent [1] and OpenHands [2] which are not evaluated here.
- The experiments also use a temperature of 0 which does not necessarily guarantee determinism for modern LLMs.
- There is no justification provided for the choice of security impacts. Is there a concrete definition or list of security impacts that can be cited here?
- While I appreciate the list of insights from the experiments in Section 4.2, I do not understand how this guides (a) practioners who would like to use LLMs for this task and (b) future research in this area. The proposed future directions in Section 6 are applicable to the use of LLMs for most tasks, are there any pointers specific to the studied task?

There are some instances in the main text that should ideally include citations. For instance, line 136 mentions "Recent researchers show that many labeled vulnerable functions..." but this is not supported with a reference. Similarly, line 098 mentions that the benchmark "covers the three main categories of security impacts" but this isn't supported with a reference / justification.

Overall, I think the paper would immensely benefit from including state-of-the-art LLM-based agents in the evaluation and listing actionable insights which can guide either practioners / researchers in this area. Further, supporting claims with references should improve the exposition of the text.


[1] Yang, J., Jimenez, C. E., Wettig, A., Lieret, K., Yao, S., Narasimhan, K., & Press, O. (2024). Swe-agent: Agent-computer interfaces enable automated software engineering. Advances in Neural Information Processing Systems, 37, 50528-50652.

[2] Wang, X., Li, B., Song, Y., Xu, F. F., Tang, X., Zhuge, M., ... & Neubig, G. (2024). Openhands: An open platform for ai software developers as generalist agents. arXiv preprint arXiv:2407.16741.

**Questions:**

I will summarize my questions from the weaknesses section above (please refer to that section for more details):
1. Could you provide a list of actionable insights which can guide practioners / future researchers in this area?
2. Could you comment on the exclusion of SotA LLM-based frameworks from the evaluation?
3. How were the three security impacts chosen?
4. How well does the temperature = 0 ensure determinism of outputs?

Additionally,

5. How many samples are there in the benchmark? Line 218 mentions that SecTest-Eval includes 204 instances while Table 1 states a total of 203.

---

> ### Author Response · Authors · 2025-12-04
> **Response to Reviewer Q1vV**
>
> We thank Reviewer Q1vV for acknowledging that our benchmark is the first to look at the*security impacts of function-level vulnerabilities and that our experiments section is well-written and easy to understand.
>
> > Weakness: Exclusion of SOTA LLM-based agentic frameworks.
>
> We agree that evaluating LLMs equipped with agents in our evaluation task can capture the frontier of existing methods.
>
> We have evaluated 3 state-of-the-art agents, including agents for security tasks like Enigma, and agents for general software engineering tasks like SWE-Agent and OpendHands.
>
> We report the experiment results in the latest version of paper, which shows that agent integration improve success rate by 7.5% to 63.5%, indicating equipping LLMs with tools can improve performance
>
> > Weakness: No justification for the choice of security impacts.
>
> We have clarified this in the SecTest-Eval Section in the latest version of paper.
>
> The three security impacts types formulated by us, including Unauthorized Data Reading, Unauthorized Data Modification, and Denial-of-Service, are explicitly chosen because they align with the fundamental security properties of the basic security properties, including Confidentiality, Integrity, and Availability, which are the standard, foundational elements used to define security requirements across the industry. Furthermore, these types cover the most prevalent weaknesses in the 2024 CWE Top 25 list.
>
> > Weakness: How well does temperature=0 ensure determinism?
>
>  We agree that determinism must be validated and have reported the experiment results of it in the latest version of paper.
>
>  we conducted five repeated experiments using the GPT-4.1 model with direct output prompting, and the resulting standard deviation of4.7% confirms the temperature setting of 0 guarantees determinism
>
>
> > Weakness: Lack of actionable insights.
>
> We agree the importance of identify the impression of our findings.
>
> We have clearly identified it in the latest version of paper. These findings highlight current LLMs have limitations in both understanding vulnerability exploitation and reasoning in security context, suggesting that enhancing LLMs for exploit generation requires either training on specialized datasets or incorporating security-specific tools.
>
> > Minor Comments: Citation and data consistency.**
>
> We have added the necessary citations to support claims in the Introduction and Evaluation Framework in the latest version of paper. We have also ensured consistency in the task instance count 203 across the text and tables.

---

### Official Review · Reviewer_mNdz · 2025-10-31

**Soundness:** 2
**Presentation:** 2
**Contribution:** 2
**Rating:** 2
**Confidence:** 4

**Summary:**

The paper proposes a new benchmark, SecTest-Eval, to evaluate LLMs’ ability to verify the security impact caused by a weakness. Different from prior work, SEC-bench and CyberEval, SecTest-Eval include additional information to generate the proof of concept test program, such as an impact description. SecTest-Eval also covers 14 weakness types, which are more than some prior work. The paper presents some analysis of LLMs' performance on this new benchmark, indicating that room for improvement exists, and there are some patterns of failure were discussed.

**Strengths:**

+ Security vulnerabilities are important weaknesses to study and detect. Having more benchmarks would allow better evaluation of detection and analysis techniques.

**Weaknesses:**

- The evaluation is lacking a comparison with prior benchmarks. It is important to justify the motivation for another benchmark and what this new benchmark contributes to the existing landscape. And it is also very crucial to quantitatively show the uniqueness of this new benchmark compared to the existing ones.
- The comparison with the prior benchmark is not complete; there is a need for a more detailed analysis of overlap and unique entries in terms of weakness types. Also, a separability and ranking agreement analysis between TestSec_Eval and prior benchmarks, such as Sec-Bench and CyberEval, would provide a better view of how the proposed benchmark contributes to the overall landscape of existing security benchmarks.
- Lacks a proper justification for why the three primary security impacts chosen are the most important. Is there any evidence or prior study that supports this selection?

**Questions:**

1. LLMs only achieve an 18% success rate on SEC-Bench, meaning the existing benchmarks are still quite challenging. Why do we need yet another security benchmark? Why do we need to change the usage setting and provide a different set of inputs for LLMs?

2. Since SecTest-Eval covers 14 weakness types, what is the overlap with vulnerability types of prior work (memory access and privacy issues)?

3. Is there any evidence or prior study that supports the selection of three primary security impacts?

**Details Of Ethics Concerns:**

It is not clear what license the benchmark will be in. However, it is important to have a license that satisfies the license constraints from all repositories from which the data was collected. The paper does not discuss this important point.

---

> ### Author Response · Authors · 2025-12-04
> **Response to Reviewer mNdz**
>
> We thank Reviewer mNdz for the detailed feedback and for recognizing the importance of studying security vulnerabilities and the improved coverage of weakness types in our benchmark.
>
> > Weakness: Lacking comparison with prior benchmarks and justification for yet another benchmark.
>
> We appreciate your attention to the distinctions between our work and existing benchmarks.
>
> Existing benchmarks primarily evaluate LLMs from a global perspective, where LLMs are tasked to generate exploits that call vulnerable code from project entry points, and reveal significant performance gaps.
>
> Therefore, recent studies have explored decomposing the whole challenging exploit generation task into a series of relatively simple tasks, applying LLMs from a local perspective. While such attempts have shown effectiveness, existing benchmarks may lead to unreliable model performance in these scenarios due to low label accuracy for vulnerable functions. To address this, we introduce SecTest-Eval.
>
>
> > Weakness: Overlap analysis and ranking agreement with prior benchmarks.
>
> We agree on the value of a detailed overlap analysis. We will include a detailed table in the Appendix comparing the CWE types covered in SecTest-Eval with prior works.
>
> > Weakness: Lacks proper justification for why the three primary security impacts are chosen.
>
> We have clarified this in the SecTest-Eval Section in the latest version of paper.
>
> The three security impacts types formulated by us, including Unauthorized Data Reading, Unauthorized Data Modification, and Denial-of-Service, are explicitly chosen because they align with the fundamental security properties of the basic security properties, including Confidentiality, Integrity, and Availability, which are the standard, foundational elements used to define security requirements across the industry. Furthermore, these types cover the most prevalent weaknesses in the 2024 CWE Top 25 list.
>
> > Ethics Concern: License compliance.
>
> We agree that license compliance is crucial.
>
> We will explicitly state the license under which SecTest-Eval will be released and confirm that it satisfies the license constraints from all source repositories, including PrimeVul, in the final version.

---

### Official Review · Reviewer_Hagu · 2025-11-01

**Soundness:** 2
**Presentation:** 2
**Contribution:** 2
**Rating:** 2
**Confidence:** 4

**Summary:**

This paper proposes SecTest-Eval, a function-level benchmark, evaluates whether LLM can generate a good PoC test for vulnerabilities that can verify security impacts for C/C++ programs. Basically, the input for LLM is a triplet ⟨vulnerable function fv, its CWE type, and a target security impact I⟩, the model should output a self-contained unit test format program T that can be directly tested by execution and trigger the problem. The benchmark construct 203 function-level tasks, which cover 14 CVE types with 3 main impact categories. They evaluate the whole benchmark with five SOTA LLMs under "Direct" and "COT" mode prompting.

**Strengths:**

1. Clear and well-scoped formulation. The task is precisely defined ⟨vulnerable function fv, its CWE type, and a target security impact I⟩ -> T. The benchmark provides concrete, automatically verifiable oracles for each impact type—making evaluation reproducible and objective.
2. Enough experiment workload, provide an evaluation of the SOTA models and check with two kinds of prompts(direct and COT) to show the diverse results.

**Weaknesses:**

1. Limited realism of function-level scope. Generally, real vulnerabilities often span multiple functions or modules. Evaluating only isolated functions misses complex control/data-flow dependencies—meaning success on SecTest-Eval may not translate to real-world exploitation.
2. Minor contribution and limited diversity. The dataset largely comes from previous work PRIMEVUL, with additional filtering and relabeling. While this makes the benchmark easier to construct, it limits originality. Moreover, the dataset only covers C/C++ code, which may introduce bias and prevent a more comprehensive evaluation of LLM capabilities across languages and ecosystems.
3. Trivial evaluation method. The evaluation of generated PoCs mainly relies on executing the produced code and checking for explicit runtime signals (e.g., file modification or crash). This rule-based approach cannot capture more subtle or complex exploit behaviors.
4.  Limited experimental design. The experiments focus only on direct model prompting and do not consider agentic or tool-augmented settings, which are a core capability of many state-of-the-art LLM. Without including agent-based reasoning or tool use, it remains unclear whether the observed weaknesses stem from model limitations or simply from the lack of external reasoning support.

**Questions:**

1. It might be good to cite some secure code generation work as a shared related interest for LLM security.

[1] Seccodeplt: A unified platform for evaluating the security of code genai

[2] SafeGenBench: A Benchmark Framework for Security Vulnerability Detection in LLM-Generated Code

[3] CodeLMSec benchmark: Systematically evaluating and finding security vulnerabilities in black-box code language models

2. Do you consider testing with some SOTA code agent, like OpenHands, Claude-Code?

---

> ### Author Response · Authors · 2025-12-04
> **Response to Reviewer Hagu**
>
> We appreciate Reviewer Hagu for acknowledging the clear formulation, objective evaluation oracles, and the sufficient experimental workload of our paper.
>
> >Weakness: Limited realism of function-level scope.
>
> We acknowledge your focus on realism of our evaluation task, where LLMs are tasked to generate local exploits directly calling a vulnerable function.
>
> While this task seems limited realism now, it is designed to enable practical applications of LLMs in exploit generation task in the real-world. As existing work has demonstrated potential of applying LLMs in sub-tasks of the whole and complex exploit generation task. Furthemore, LLMs' potential in our evaluation task is also demonstrated by existing work like Magneto, and our work tackle the problem that existing benchmark cannot reliably evaluate LLMs in this task due to low label accuracy for vulnerable functions.
>
> >Weakness: Limited experimental design (Exclusion of agentic frameworks).
>
> We agree that evaluating LLMs equipped with agents in our evaluation task can capture the frontier of existing methods.
>
> We have evaluated 3 state-of-the-art agents, including agents for security tasks like Enigma, and agents for general software engineering tasks like SWE-Agent and OpendHands.
>
> We report the experiment results in the latest version of paper, which shows that agent integration improve success rate by 7.5% to 63.5%, indicating equipping LLMs with tools can improve performance.
>
> > Weakness: Minor contribution and limited diversity (Data largely comes from PrimeVul, C/C++ only).
>
> SecTest-Eval focus on exploring a initial step in constructing benchmarks for evaluating LLMs' capabilities in exploit generation from a local perspective, which may lack comprehensive coverage on languages, weakness types, and security impacts in current version. Besides, the design of using PrimeVul dataset is for effectiveness, one can construct task instance of our benchmark by combine automated labeling approaches in PrimeVul and ours.
>
> As stated in the Limitations and Future Work section, we will implement a labeling framework combining PrimeVul and ours in the future. Besides, the design of our framework is independent of these factors, and we plan to systematically extend it to other languages, CWEs, and impacts in future work.
>
> > Weakness: Trivial evaluation method (Rule-based checking for file modification/crash).
>
> We agree the importance of precisely reflecting LLMs capabilities in exploit generation.
>
> Following existing exploit generation benchmarks, we use evaluation metrics based on rule-based checking. However, we have a smaller input context than existing benchmark, which make the same evaluation metrics more precise.
>
> We will explore more metrics for precise evaluation in the future.
>
> > Question: Cite secure code generation work and consider SOTA code agents.
>
>  We appreciate the suggestion and have added the requested secure code generation benchmarks, including Secodeplt , SafeGenBench, and CodeLMSec in the Related Work section, where we position our work as complement evaluating LLMs in different security tasks. We have also confirmed our evaluation of state-of-the-art agents.

---

### Official Review · Reviewer_TATa · 2025-11-02

**Soundness:** 2
**Presentation:** 1
**Contribution:** 2
**Rating:** 2
**Confidence:** 5

**Summary:**

LLMs show promise in analyzing security vulnerabilities but current benchmarks may underestimate their true capabilities due to excessive context length and limited coverage of exploitation methods and security impacts. The paper introduces SecTest-Eval, a benchmark that tasks LLMs with generating PoCs, unit tests to verify security impacts of 203 vulnerable C/C++ functions across 14 weakness types and 3 impact categories. Evaluation of five state-of-the-art LLMs reveals modest success rates of up to 56%, highlighting the need for further improvement in LLMs’ ability to accurately assess security vulnerabilities.

**Strengths:**

1. Addresses evaluating LLMs' capabilities for the problem of PoC generation, an essential and time-consuming step for assessing the security impact of any vulnerability. The problem is very challenging, it is timely to solve in the era of LLMs, and it is under-studied relative to its practical importance.

2. Proposes a novel benchmark for generating PoCs in the form of programs containing unit tests that are specifically designed to validate the security impacts of a given vulnerable function.

3. Curates a balanced dataset of 203 samples, each consisting of a C/C++ vulnerable function, the type of the contained weakness (across 14 types of CWEs), and 3 impact categories (Unauthorized Data Reading, Unauthorized Data Modification, and Denial of Service).

4. Demonstrates headroom for improving LLM's abilities to generate PoCs, with only 56% overall accuracy for GPT-4.1.

**Weaknesses:**

1. The benchmark is limited to function-level vulnerabilities. The authors consider this a strength since the latter can underestimate the true capabilities of LLMs due to excessive context length, but it is also a significant weakness to focus on single functions.

2. The algorithm for how PoCs are generated is not presented. This is a lost opportunity since it would be essential to understanding important aspects of your approach such as its effectiveness (e.g. in terms of running time, LLM inference cost, and any guarantees it provides) and generality (e.g. in terms of extending it to support multiple functions, and different kinds of CWEs and security impacts). Ideally, the presentation of the algorithm should be in terms of the formal notation introduced in the problem formulation.

3. The paper has other presentation issues. Most importantly, there is no example of a generated PoC. The appendix shows many examples of non-PoCs but not a single example of a valid PoC. I would expect at least one such example, ideally in the main body of the paper, and an illustration of how the method generates it.

4. The paper claims generality as a strength over existing benchmarks in terms of aspects such as coverage of exploitation methods and security impacts. But it focusses somewhat narrowly on C/C++, and a few CWE types and impact categories.

Minor comments:

- Figure 3 is not referenced at all, making definitions 1-3 hard to understand.
- I could not find statistics of valid PoCs such as lines of code. This would help in getting a sense of how complex PoCs your framework can generate.
- A start of Section 3, you say 204 task instances, but Table 1 says 203.

**Questions:**

Please see weaknesses.

---

> ### Author Response · Authors · 2025-12-04
> **Response to Reviewer TATa**
>
> We thank Reviewer TATa for recognizing the timeliness, novelty, and practical importance*of our work.
>
> >Weakness: Limited to function-level vulnerabilities.
>
> We agree that real-world vulnerabilities can span multiple functions.
>
> Following existing work which formulates constructing exploits directly calling a single function to trigger security impacts, we focus on single-function vulnerabilities in current SecTest-Eval.
>
> While our evaluation shows generating exploits for a single vulnerable function still challenges LLMs, even equipped with agents, we plan to address exploit generation for multi-function vulnerabilities in the future.
>
> >Weakness: Algorithm for PoC generation is not presented.
> The algorithm of LLM code generation task can be described by model input and output due to models' unknown internal activities.
>
> In SecTest-Eval, the model input is a vulnerable function, a description of its weakness and a description of the target security impact, and the model output is an exploit program directly calling the vulnerable function.
>
> In the future, we will explore the internal 'algorithm' of LLMs when they handle exploit generation. A possible direction is similar to this study, which divides the whole exploit generation task into sub-tasks and evaluates LLMs in each sub-task.
>
> >Weakness: No example of a generated PoC.
>
> We agree this is a critical missing piece. We have added analysis for examples of generated exploits, providing a side-by-side comparison of a successful exploit generated by GPT-4.1 and a failed exploit generated by Qwen3-Coder.
>
> >Weakness: Narrow focus on C/C++, few CWE types and impact categories.
>
> SecTest-Eval focus on exploring a initial step in constructing benchmarks for evaluating LLMs' capabilities in exploit generation from a local perspective, which may lack comprehensive coverage on languages, weakness types, and security impacts in current version.
>
> As stated in the Limitations and Future Work section, the design of our framework is independent of these factors, and we plan to systematically extend it to other languages, CWEs, and impacts in future work.
>
> >Minor Comments: PoC statistics, figure references, and data consistency.
>
>  We have included the requested statistics in the revised paper, showing average lines of generated exploits and the iterative steps of LLMs. We have also ensured consistency in the task instance count 203 and explicitly referenced Figure 3.

---

### Meta-Review · Area_Chair_GjCr · 2026-01-09

**Summary:**

All reviewers agree that the paper targets a timely and practically important problem—evaluating LLMs for vulnerability impact verification via PoC-like unit tests—and they recognize that SecTest-Eval provides a clear formulation and reproducible evaluation setup. However, the decision is driven by a consistent set of substantive concerns: the benchmark’s limited realism and generality, due to its focus on function-level vulnerabilities and a relatively narrow scope (C/C++ only, limited CWE/impact taxonomy); the insufficient justification of novelty relative to existing benchmarks and lack of rigorous quantitative comparison (e.g., overlap, separability, ranking agreement); and the incompleteness of experimental methodology, especially in the original version where evaluations relied primarily on simple prompting and a relatively rule-based notion of success that may not capture nuanced exploit behaviors. In addition, reviewers raised significant presentation and clarity gaps, including missing concrete examples of valid PoCs, missing or inconsistent dataset counts, and underdeveloped explanations of how tasks are generated and why specific security impacts were chosen. Overall, while the benchmark is promising, reviewers found the contribution not mature enough for acceptance in its current form.

**Reviewer Concerns:**

The rebuttal partially addressed several key issues, including adding (i) at least one example of successful vs. failed exploits, (ii) PoC statistics and consistency fixes for task counts, (iii) additional citations and references, (iv) an explicit rationale for the three security impact categories through CIA (confidentiality/integrity/availability) alignment and prevalence in CWE lists, and (v) new experimental results incorporating agentic frameworks (SWE-Agent, OpenHands, Enigma), which strengthens the experimental story and mitigates the earlier criticism of limited evaluation design. That said, important concerns remain outstanding: the benchmark’s central limitation—function-level scope—continues to raise questions about how well success transfers to real-world vulnerabilities, and the rebuttal largely positions extensions as future work rather than convincingly demonstrating current realism. Likewise, while the authors explain why another benchmark is needed, the response still does not provide a sufficiently deep quantitative comparison with prior benchmarks (e.g., overlap analysis, uniqueness metrics, agreement analysis) that would justify SecTest-Eval as a distinct and necessary addition to the landscape. Finally, the discussion of the PoC-generation “algorithm” remains limited, and the evaluation methodology still relies heavily on relatively coarse runtime signals that may miss more subtle exploit impact behaviors. Thus, while the rebuttal improves clarity and breadth, it does not fully resolve the main concerns driving rejection.

**Reviewer Scores:**

Reviewer TATa would likely remain at Reject (2), as their primary concerns—function-level limitation, missing algorithmic details, and missing strong PoC examples—are only partially mitigated; while the rebuttal adds examples and minor fixes, the lack of a concrete PoC generation process and broader generality likely keeps their confidence in rejection high. Reviewer Hagu might slightly soften their stance (e.g., from a firm Reject to a borderline Reject), but would still likely remain at Reject (2) because although agent-based evaluations were added, their critiques about limited realism, limited originality due to dependence on PRIMEVUL, and rule-based evaluation remain largely unaddressed at a fundamental level. Reviewer mNdz might also shift slightly upward in sentiment due to the clarified motivation and the CIA-based justification of impact categories, but would likely stay at Reject (2) because the rebuttal still lacks the kind of quantitative benchmarking comparison and overlap/separability analysis they explicitly requested, and the ethics/licensing issue, while acknowledged, remains prospective rather than fully resolved. Reviewer Q1vV may have improved somewhat given the inclusion of agents, determinism testing, citations, and clearer justification of impact categories, but would also likely remain at Reject (2) since concerns about contribution strength, actionable insights, and the extent to which the benchmark advances practice remain only partially satisfied. Overall, the rebuttal improved the paper, but likely not enough to convert any reviewer to an accept stance in discussion.

---

### Decision · Program_Chairs · 2026-01-26

Reject